# Lead Recovery from Solid Residues of Copper Industry Using Triethylenetetramine Solution

**Mateusz Ciszewski \***[iD]**, Andrzej Chmielarz, Zbigniew Szołomicki, Michał Drzazga** **and Katarzyna Leszczyńska-Sejda**[iD]

ŁUKASIEWICZ—Institute of Non-Ferrous Metals, Sowińskiego 5, 44-100 Gliwice, Poland; andrzejch@imn.gliwice.pl (A.C.); zbigniewsz@imn.gliwice.pl (Z.S.); michald@imn.gliwice.pl (M.D.); kasial@imn.gliwice.pl (K.L.-S.)

\* Correspondence: mateuszc@imn.gliwice.pl; Tel.: +48-32-23-80-277

**Abstract:** Industrial processing of mineral ores and concentrates generates large amounts of solid residues, which can be landfilled or further processed to recover selected elements depending on its economical profitability. Pressure leaching is a technology enabling high recovery of base metals like copper and zinc, transferring others like lead and iron to the solid residue. High temperature and pressure of such leaching leads to formation of sparingly soluble lead jarosite (plumbojarosite). The load of lead landfilled as solid residues resulting from such operation is so big that its recovery is perspective and crucial for waste-limiting technologies. This paper is devoted to lead extraction from pressure leaching residues using triethylenetetramine solution and then its precipitation as a commercial lead carbonate. The highest obtained recovery of lead was 91.3%. Additionally, presented technology allows to manage and recycle amine solution and reuse solid products. Produced pure lead carbonate can be directly added to smelting, not increasing temperature within the furnace.

**Keywords:** triethylenetetramine; lead extraction; recycling; lead carbonate

## 1. Introduction

Increasing metal demands and decreasing economic grades of copper deposits lead to seeking alternative routes to pyrometallurgical copper processing [1]. Historically, the pyrometallurgical processing of ores dominated the copper industry since the 1800s, however, since the 1990s, the increase share of hydroprocessing plants can be observed [2]. The advantages of hydrometallurgical technique include high metal recovery, greater tolerance towards impurities, flexibility, modularity, lower energy consumption, and lower capital costs of launching and commissioning of plants, however, limitations like lower productivity and large amounts of generated effluents and residues have to be taken into account too. An increase in leaching efficiency can be obtained using more severe process conditions i.e., high temperature and pressure. Pressure leaching is based on acidic dissolution of the sulfide matrix at high temperature (at least 190 °C) and pressure (20 bars total) [3]. Depending on the acidity and the oxygen amount, the various iron compounds may be produced, either hematite (T < 200 °C, low acidity, reaction 1), basic iron sulfate (160 °C < T < 200 °C, high acidity, reaction 2), or jarosites (high acidity and presence of $K^+$, $NH_4^+$, $Ag^+$, $Pb^{2+}$, reaction 3 and 4) [4,5]. Presence of the specific metal cations ($Pb^{2+}$, $Na^+$, $K^+$, $Rb^+$, $Ag^+$ etc.) within the metal ore or concentrate may results in different jarosites formation.

$$Fe_2(SO_4)_3 + 3H_2O = Fe_2O_3 + 3H_2SO_4 \text{ (hematite)} \tag{1}$$

$$Fe_2(SO_4)_3 + 2H_2O = Fe(OH)SO_4 + 8H_2SO_4 \text{ (oxidation + hydrolysis)} \tag{2}$$

$$3Fe_2(SO_4)_3 + 14H_2O = 2H_3OFe_3(SO_4)_2(OH)_6 \text{ (hydronium jarosite)} \tag{3}$$

$$3Fe_2(SO_4)_3 + M_2SO_4 + 12H_2O = 2\,MFe_3(SO_4)_2(OH)_6 + 6H_2SO_4 \qquad (4)$$

where $M = Ag^+$, $NH_4^+$, $K^+$, $1/2Pb^{2+}$.

Formation of jarosite and basic iron sulfates are aimed to immobilize and remove iron from the reaction system. However, these compounds make environmental and processing problems in the future as well, as they both pose difficulties in further recovery of silver from the leaching residues [6]. There have been made several attempts to decompose yet produced jarosites using sulfuric acid [7], hydrochloric acid with an addition of calcium chloride [8], ammonia [9], and sodium or calcium hydroxides [10,11]. Among them, sulfuric acid seems to be the most convenient regarding further metal treatment as it is less corrosive and troublesome (vapours) in comparison to ammonia or hydrochloride. Additionally, for the proposed technology using triethylenetetramine solution, there is no need to make a next unit operation for chloride removal as sulfate ions can be "recovered" in a gypsum production stage [12–14]. Lead from decomposed lead jarosite can be then extracted using various techniques, mostly used for lead-acid battery paste desulfurization. In fact, desulfurization of battery paste containing lead sulfate and lead oxides leads to metal lead recovery [15,16]. Several hydrometallurgical attempts can be distinguished in lead sulfate processing with the use of sodium hydroxide, sodium carbonate, ammonium acetate, and citric acid among the most popular [17]. Depending on the stoichiometry of lead sulfate to sodium carbonate, the reaction products can be lead carbonate ($PbCO_3$), lead hydroxycarbonate ($Pb_3(CO_3)_2(OH)_2$), or sodium lead hydroxycarbonate ($NaPb_2(CO_3)_2OH$) and other intermediates [18]. Desulfurization with sodium carbonate or more troublesome sodium hydroxide generally exceeds 99% [19]. In both cases, up to 10% excess of desulfurizing agent has to be used. Another approach for spent lead acid battery processing uses ammonium acetate, however, it requires thermal pretreatment of lead paste to oxidize metal impurities (Ca, Fe, Sb, Ba) and then sulfuric acid to produce soluble metal sulfates [20]. Phase separation of as obtained material allows to leach lead sulfate with ammonium acetate. Efficient desulfurization can be done using a mixture of sodium citrate and citric acid, resulting in lead citrate and sodium sulfate [21]. Decomposition of lead oxides (PbO and $PbO_2$) requires excess of citric acid and addition of hydrogen peroxide. The perspective method to recover metal constituents from various solid materials like sludges, fly ashes (coal fly ash, oil-fired fly ash, municipal wastes incineration fly ash), electronic wastes, spent catalysts, as well as low grade ores and secondary wastes, is bioleaching [22]. This approach used with a proper pretreatment (catalyst, ultrasound, grinding) allows to significantly enhance metal extraction. Here, we reported a novel approach to extract lead from pressure leaching residue priorly treated with sulfuric acid solution. Triethylenetetramine solution was used to extract lead sulfate, which was then carbonated to produce pure lead carbonate. The biggest advantage of the proposed methodology is the possibility to recycle the extracting solution, minimizing chemical consumption and environmental impact [23].

## 2. Materials and Methods

Solid residues were obtained in pressure leaching of copper concentrates delivered from three different European copper mines (localized in Serbia—SE, Poland—PL, Portugal—OR). The main host mineral for copper was covellite (SE), chalcopyrite and bornite (PL), and chalcopyrite (OR). Copper concentration in polymetallic concentrates varied from 1.7% (SE), through 5.2% (OR) to 13% (PL). Processes were carried out at 190–200 °C with an oxygen overpressure 5–7 bars in a reactor equipped with heating jacket, cooling coils, baffles, temperature and pressure measurement instruments, gas feed valves and controllers, and mechanic agitator. Produced solid residue was filtered, washed with water and dried. In the pre-treatment, as-obtained solid residues were mixed with 1 M sulfuric acid solution at temperature 90 °C to decompose lead jarosite and produce lead sulfate.

Resulting sludge was filtered and washed with water to remove free acid (confirmed by a pH test) and then treated with 6% *w/w* triethylenetetramine solution in water. Extraction of lead sulfate with amine was carried out for 1 h at 60 °C with the solid:liquid ratio 0.2. As triethylenetetramine is a viscous liquid, an increased temperature (60 °C)

was used to enhance wettability and to improve metal salt extraction. pH was monitored within the process. It dropped from 11.5 for the fresh water-amine solution to 10 after an hour. A longer time did not cause any significant pH change, which may be ascribed to the end of the extraction process. Phases were then separated on Buchner funnel and filter cake was rinsed with water to remove any occluded amine droplets rich in lead sulfate. Filtrates and washings were then mixed and carbonated using specially-designed teflon-lined flotation machine equipped with agitator, stator, and gas nozzles enabling to deliver gas to the bottom of the reaction vessel and uniformly distributed it in a whole volume to achieve its high dissolution. Purging of carbon dioxide through lead sulfate containing triethylenetetramine solution led to precipitation of sparingly soluble lead carbonate [24]. When pH of solution was close to 7, the lucidity turned from clear to milky. Carbonation was carried out till pH 6.5, as it was found that at pH 7 there was still around 0.5 g Pb/L while at pH 6.5 the Pb level was below 0.1 g/L. Vacuum filtration was used to separate lead carbonate. Triethylenetetramine solution containing sulfate ions could then regenerated by removal of sulfate ions. This could be done using the simple precipitation method with addition of lime and producing gypsum ($CaSO_4 \cdot 2H_2O$). As obtained amine solution with depleted sulfate ions could once again used in lead sulfate extraction.

Qualitative and quantitative analysis of all materials streams were performed using spectrometer Panalytical Axios Max, flame atomic absorption spectroscopy (FAAS) with iCE3300 from Thermo Fischer Scientific (Schwerte, Germany), titration methods, and X-ray powder diffraction using Rigaku MiniFlex 600 (Wrocław, Poland)with Cu Kα radiation, equipped with silicon strip detector D/teX, and Soller slits 2.5".

Chemical reagents of a grade pure for analysis: triethylenetetramine (60% solution, across organics), sulfuric acid (96%, POCh), iron(III) sulfate pentahydrate—as additional oxidant for non-ferrous metal leaching (chempur), calcium hydroxide (chempur), oxygen, and carbon dioxide (Linde) were used.

## 3. Results

Quantitative analysis of pressure leaching residues produced in copper-bearing concentrates processing revealed variation in lead content from about 6 up to 20 wt% (Table 1). Zinc and copper, which can be co-extracted with lead, were present in a minor extent. Sulfur speciation showed that it was present in sulfate form. This is quite important as triethylenetetramine is very aggressive towards copper and is able to react with sulfur sulfide as well. Therefore, a reduced amount of some selected elements is desirable.

**Table 1.** Elemental composition of pressure leaching residues prepared from various copper concentrates.

| Sample | Pb | Zn | Cu | Fe | As | Al | Si | Ag | $S^{TOT}$ | $S^{SO4}$ |
|--------|------|------|------|-------|------|------|-------|-------|------|------|
| SE | 6.68 | 0.07 | 0.05 | 21.1 | 0.25 | 0.07 | 1.41 | 0.006 | 13.5 | 13.32 |
| PL | 6.37 | 0.05 | 1.25 | 10.4 | 0.54 | 4.72 | 10.95 | 0.077 | 6.75 | 6.61 |
| OR | 19.7 | 0.31 | 0.31 | 29.45 | 0.21 | 0.10 | 0.38 | 0.029 | 11.75 | 10.97 |

The XRD patterns all types of analyzed materials showed plumbo-jarosite ($PbFe_6(OH)_{12}(SO_4)_4$) as a dominating phase, signals attributed to anglesite ($PbSO_4$), quartz ($SiO_2$), and anhydrite ($CaSO_4$) with very high intensity signal for sample PL (Figure 1). Except those signals, additionally, in sample SE peaks ascribed to barite ($BaSO_4$) and iron hydroxide sulfate ($FeOHSO_4$) were found. Pressure leaching of copper concentrate, conducted at low pH, high temperature, and pressure immobilizes practically all lead in the form of sparingly soluble plumbo-jarosite. Therefore, it is obligatory to decompose it into easy processable lead salt.

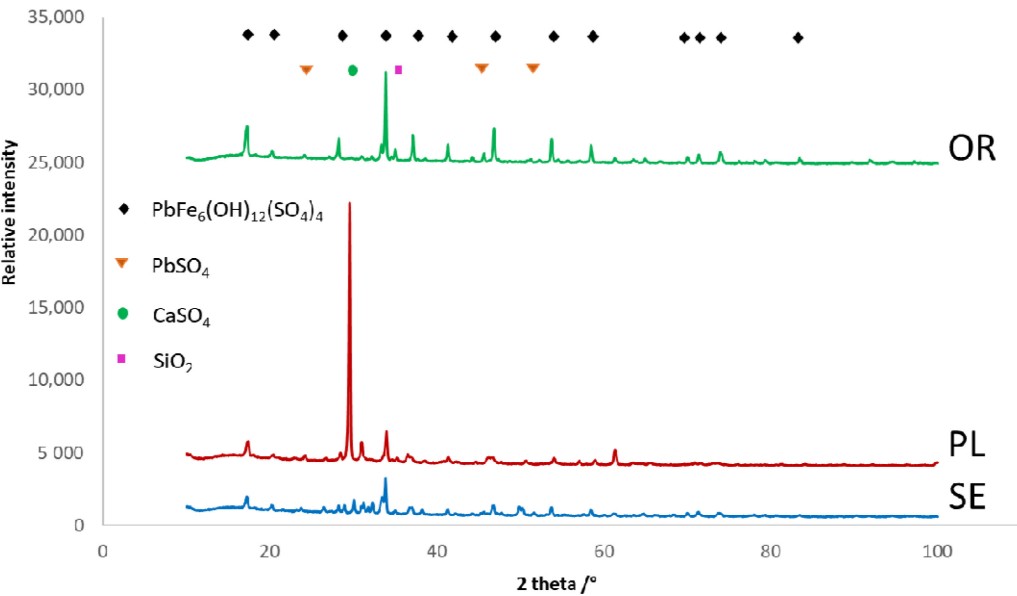

**Figure 1.** XRD patterns for pressure leaching residue.

Decomposition of plumbo-jarosite was performed in 1 M sulfuric acid solution for 20 h at 90 °C with an intensive mechanical mixing. This procedure was applied for all three samples and resulted in total conversion of plumbo-jarosite into lead sulfate. After the acid treatment concentration of lead within the examined samples, there was an increase from 6.68% to 14.8 (SE), from 6.37% to 8.9% (PL), 19.7% to 59.2% (OR). Significant increase in lead concentration in OR was combined with drastic depletion of iron from 29.45% to 1.7% leached to the solution, and consequently there was big mass loss. Similarly, final concentration of iron ~1.8% was found for two PL and SE. As obtained solid materials were then treated with triethylenetetramine solution to extract lead sulfate into the liquid phase. The mechanism of lead sulfate extraction assumes formation of the complex compound of amine with lead cation while sulfate anions accompany this process (Figure 2).

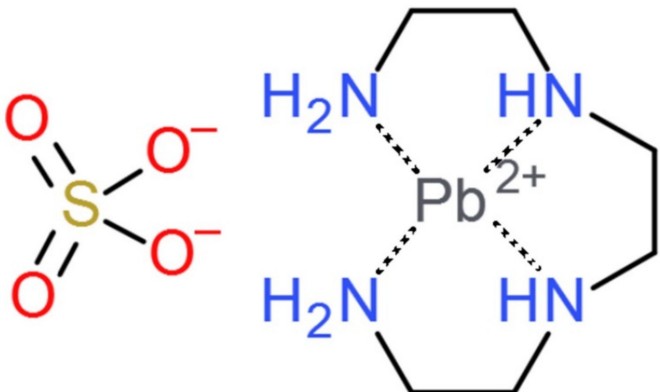

**Figure 2.** Complex of triethylenetetramine with lead sulfate molecule.

Table 2 summarizes data obtained within the leaching process of pressure leaching residues pre-treated with sulfuric acid solutions. The highest extraction was obtained for material SE (91.3%), slightly smaller for OR, and finally, most disappointing results were found for PL. Although, mass losses for sample PL and OR may indicate that all lead was extracted, in fact, big depletion of mass was caused by co-extraction of anhydrite from the PL sample. Some of the two valence metal cations are very prone for being extracted by aliphatic amines and calcium is one. Mass loss equals 29.3%, while the anglesite content in PL is merely around 13%.

**Table 2.** Parameters of the lead sulfate extraction in triethylenetetramine solution.

| Sample | SE | PL | OR |
|---|---|---|---|
| $C_{Pb}^{SOLID}/\%$ | 14.8 | 8.9 | 59.2 |
| $pH^{TETA}$ start | 12 | 11.8 | 12 |
| $pH^{TETA}$ final | 11.1 | 11.0 | 11 |
| Mass loss/% | 27.3 | 29.3 | 76.4 |
| Pb leached/% | 91.3 | 37.8 | 86.2 |

Anglesite-rich amine solution was then treated with carbon dioxide with the volumetric flow rate 50–70 L/h. Gas was delivered through the specially designed mixer and injected under the blades in a close vicinity to the bottom of reactor. Intensive mixing allows uniform distribution of carbon dioxide molecules, which were reacted with lead present in the amine solution. As a result of this reaction, a sparingly soluble lead carbonate was precipitated, oxygen gas was evolved from the mixture, and sulfate anions were left in the triethylenetetramine solution. Precipitation was carried out until pH of mixture was about 6.5. Phase separation and washing allowed to produce cerrusite (lead carbonate) of high purity (Table 3).

**Table 3.** Purity of produced precipitated cerrusite.

| Sample | Pb/% | Fe/% | Cu/% | As/% | Ni/% | Wettness/% |
|---|---|---|---|---|---|---|
| SE | 76 | <0.005 | <0.0025 | <0.01 | 0 | 20 |
| PL | 75.7 | 0.25 | <0.0025 | <0.01 | 0.03 | 24 |
| OR | 75.8 | <0.005 | <0.0025 | <0.01 | 0 | 22 |

Morphological characterization of the obtained lead carbonate was performed using SEM technique. It demonstrated that the petal-like structures were crystallized. Additionally, images recorded magnification 10 and 100 kX showed that the obtained product was in nanoscale (Figure 3). This can be assigned to high stress forces of the mixer's blades and uniform dispersion of gas within liquid, satisfying non-disruptive precipitation. Very similar images were obtained for all three cerrusite samples.

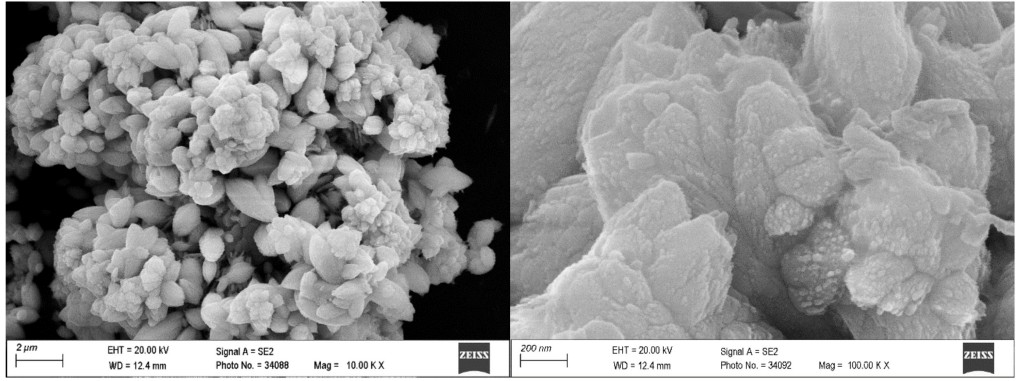

**Figure 3.** SEM images of lead carbonate precipitated from triethylenetetramine solution.

This technology allows to precipitate lead into the trade product without supply of any other chemical agents (metal ions) into the solution. The injected carbon dioxide produces pure precipitate and leaves only sulfate ions in triethylenetetramine solution. However, any calcium sulfate that is coextracted with lead sulfate is obviously precipitated by carbon dioxide to calcium carbonate. Therefore, it is recommended to remove any interfering ions (like calcium) prior to amine, unless strength of metal (other than Pb)-amine bond is bigger than lead-amine.

To obtain triethylenetetramine solution that can be recycled to the extraction stage, it is recommended to remove all sulfate ions. It is suggested to use lime suspension $(Ca(OH)_2)$. It can be either lime in water or lime in amine solution, depending on the water balance of the process. Lime has to be added in such amount to precipitate all sulfate producing calcium sulfate (in a hydrated form) and limiting any excess to have amine solution without any calcium ions. Produced gypsum $(Ca(OH)_2 \cdot 2H_2O)$ can be either landfilled or sold as a market product, when its purity, size of crystals (length to width ratio), and wetness are appropriate. The quality of produced calcium sulfate to hydrate with respect to metal contents was quantitatively analyzed and presented in a Table 4.

**Table 4.** Concentration of impurities of the produced gypsum (in % *w/w*).

| Sample | Pb/% | Fe/% | Cu/% | As/% | Co/% | CaCO₃/% | Wett./% |
|---|---|---|---|---|---|---|---|
| SE | <0.0042 | 0.012 | <0.0025 | <0.01 | <0.0025 | 0.8 | 9.4 |
| PL | <0.003 | 0.012 | <0.0025 | <0.01 | <0.0025 | 0.89 | 15 |
| OR | <0.0085 | 0.011 | <0.0025 | <0.01 | <0.0025 | 1 | 17 |

It was shown that purity of all produced gypsum samples was high. However, it is recommended for commercial gypsum to have a humidity content up to 12–15%. Regarding this parameter, only one produced material fulfilled these requirements. Calcium carbonate content was similar in all samples which means that almost all calcium co-extracted with lead for PL material was co-precipitated with cerrusite. Triethylenetetramine solution, which was below 0.1 g/L Pb and 0.01 g/L Ca can be recycled and returned to the extraction stage.

## 4. Conclusions

This paper reports the possibility to use triethylenetetramine solution to extract lead salt from pressure leaching residues, which were produced in acid leaching of different copper concentrates. It was found that aliphatic amine is a powerful extracting agent and allows to extract up to 91.3% Pb in the form of anglesite. Lead can be then retrieved from the solution using simple precipitation with carbon dioxide. This results in pure lead carbonate which can be smelted. It was also proposed to recycle amine solution by an appropriate treatment and use it in the next extraction stage. It has to be emphasized that triethylenetetramine water solution is a viscous liquid and has to be washed from all solid products described in this methodology to avoid its losses.

**Author Contributions:** Conceptualization, A.C. and M.C.; methodology, M.C.; formal analysis, M.D. and K.L.-S.; investigation, M.C.; resources, Z.S.; data curation, A.C.; writing—original draft preparation, M.C.; writing—review and editing, M.C. and M.D.; visualization, A.C.; supervision, Z.S.; project administration, A.C.; funding acquisition, A.C. All authors have read and agreed to the published version of the manuscript.

**Funding:** This work was funded by Horizon 2020 project, "IntMet", Integrated, Innovative Metallurgical System to Benefit Efficiency Polymetallic, Complex and Low Grade Ores and Concentrates. Contract No. 689515.

**Informed Consent Statement:** Informed consent was obtained from all subjects involved in the study.

**Data Availability Statement:** The data presented in this study are available in Chmielarz (2014) [23].

**Acknowledgments:** Authors deeply appreciate contribution of Łukasz Hawełek and Katarzyna Bilewska in evaluation of X-ray powder diffraction results. Recognition is also due to Patrycja Kowalik for her input in lab experiments. Authors would also like to thank Witold Kurylak for project coordination.

**Conflicts of Interest:** The authors declare no conflict of interest.

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
