# Peer review of "Lead Recovery from Solid Residues of Copper Industry Using Triethylenetetramine Solution"

_minerals, doi:10.3390/min11050546_

Round 1
Reviewer 1 Report
attached

Author Response
Responses to Rev 1
„Thank you for submitting this manuscript for review, I have enjoyed reading through your work and
learning about the new suggested approach to Pb leaching from waste mineral residues.
I am providing a few comments and recommendations which will help to improve your work and
make your conclusions sound better justified.
English language
Generally, you have used plain English language which will be well understood by most
readers. Below i provide a few suggestions to correct typos and some minor rewording
which could improve your draft:
p.1 line 11- “...and pressure of such a leaching”
p.2 lines 57-“Desulphurisation with sodium carbonate or sodium hydroxide generally 57 exceeds 99% [14].
Process can be carried out with comparable yield using more troublesome sodium hydroxide.” – the second
sentence isn’t needed, just include adjective “more troublesome” in the first sentence.
p.3 line 118 “...in sulfate form.”
p.3 line 120 “...with sulfur sulfide...”
p.3 table 1 You are not explaining the reason for abbreviations SE, PL and OR for your samples.
Please either explain or replace it with A, B and C or I, II, and III so reader is not looking for
references in the text.
p.4. line 129 “...high T&p...” - please replace with words – “temperature and pressure”
p.5, line 161-162 “...oxygen molecules were gas was evolved from the mixture, and sulfate anions were left
attached to in the...”
p.6 line 197 “...can be recycled and returned to...” „
ANSWER: all mentioned errors were corrected (changes are tracked and highlighted)
„Content
Generally, the scientific hypothesis, experiment results and conclusions seem sound apart
from the assumption about sulfate ions being bonded to the triethylamine after Pb
extraction (see detailed comments below). Also, the manuscript would benefit from
providing clear information on the type of original ores used for pressure leaching.
p.1, line 32 – “Depending on the presence of certain metal cations...” – presence where?”
ANSWER: This part was slightly changed. Generally the presence of specific metal cations like sodium, potassium, lead etc. May result in different type of jarosite formation.
p.2, line 48 “the most convenient” – on what way? Most efficient? What are the numbers? + please
provide references.
ANSWER: Sulfuric acid is most convenient in jarosite treatment. As it is less corrosive nad troublesome (vapours) as ammonia or hydrochloride. Additionally, for the proposed technology with triethylenetetramine solution there’s no need to make a next unit operation for chloride removal as sulfate ions can be „recovered” in gypsum production stage.
P.2 lines 50-51- references are needed for desulfurization technique and consequent metal recovery.
ANSWER: This was added as ref 12 and 13
p.2 line 74-76: “The biggest advantage of the proposed methodology is possibility to recycle extracting
solution, minimizing chemical consumption and environ-75 mental impact [18].” – I was not able to access this
reference, but is it not your own work which you presented as a poster/talk and now put into this paper? If so
than I would recommend consulting your Editor to check if you should use this reference. You are claiming that
this draft is the original research but by providing the reference to the proceedings you are implying that these
results were at least partly published previously.
ANSWER: We strongly disagree. Triethylenetetramine solution was used in the past in our Institute for desulphurisation of battery paste. This technology was developed and patented, and then international presentation was made about its developers (Authors of this manuscript: Chmielarz&Szołomicki). After some time we have realized that similar procedure can be used for extraction of lead from hydrometallurgical wastes. And this draft is the original paper using triethylenetetramine solution which was used in the past by Us but for entirely different aim.
p.2, line 78 “Solid residues were obtained in pressure leaching processes...” of what ores?
ANSWER: We have specified only the countries from which the ores come from, however, we have no permission to use company names.
p.2 line 81 “feed material” – what are they?
ANSWER: As above
p.2 line 85 “...to remove free acid” – how the total removal of free acid was confirmed?
ANSWER: filtrate and washings were checked with pH paper. That’s all. Threfore we did not include this information in the manuscript.
p.2 line 87 “solid:liquid ration 0.2” – this means for each 1 kg of solid you used 200g of liquid – is it correct?
Seems like not much liquid to create a slurry.
ANSWER: In fact S/L = 0.2 is 200g/1L(1kg).
p.2 line 88 “an increased temperature was used to enhance wettability” – what temperature?
ANSWER: Information added
- 3 line 89 pH “dropped from 11.5 ... to 10 after an hour” – this could be due to not fully removed free acid
(see comment above).
ANSWER: Partially because of acid partially because of extraction progres
p.3 line 91 “...which may be ascribed to the end of extraction process” – is it possible that this was a
limitation of extraction due to viscosity and actually “adjusting” the solid:liquid ratio resulted in
further extraction?
p.3 line 92 – “Solid liquid ratio was adjusted” – in what way by adding triethylamine solution or just
water? To what ratio it was adjusted?
ANSWER: this part was changed
p.3 line 100 “When pH of solution was close to 7” – what was the initial pH? I don’t think it remained
10 after you washed it from solids? It is the values in table 2? Then include reference to them in the
text.
ANSWER: Corrected. After TETA leaching the pH was like 10, then dropped to 6.5-7 after carbonation.
p.3 line 101 “Carbonation was carried out till pH 6.5.” – why till 6.5?
ANSWER: we have checked that at pH 7 there is still some 0.5g/L of Pb in the solution, while at pH 6.5 the lead level was below 0.1g/L.
p.3 line 113- what iron sulfate was used for?
ANSWER: Corrected. Iron(III) sulfate pentahydrate
p.3 line 116 – “Quantitative analysis of obtained residues...” – what residues?
ANSWER: Corrected. It deals with pressure leaching residues.
p.3 lines 124-127 – what are the reasons for Ca and Ba are not presented in Table 1?
ANSWER: For TETA technology calcium is not a problem as it may be precipitated in gypsym production unit. Nevertheless its concentration was below 1%. Barium is relatively inert for TETA as well.
p.4, line 135 – “resulted in total conversion” – how the total conversion was confirmed? I would not
call this conversion but “Fe extraction”, to say that due to the Fe being extracted by acid treatment
the ratio of the Pb to the other components in the solid residue increased to ...
At the moment your text implies sudden increase in concentration of Pb, but there was no additional
source of Pb, so the concentration increased due to other components being removed.
ANSWER: tht’s true. But we are not interested in increase of lead contenet. As you properly found its rising because the mass of sample is diminishing. We are more interested to transform lead jarosite into lead sulfate. This lead sulfate and no lead jarosite were confirmed by XRD method.
- 4 line 154 - Table 2:
PL value is 8,9%, and line 137 states its 8,8%. Please correct either.
ANSWER: Corrected
Your mass loss is quite large. For example, for sample SE: you extracted 91% of Pb which
should be responsible for 15.3g mass loos, but you lost 27.3 (13.8g more)
ANSWER: As above.
p.5 line 165 – how you know this is not PbS but PbCO3?
ANSWER: XRD diffraction patterns confirmed cerrusite formation.
p.6-176-179 how can sulfate ions the combined (you probably meant “bonded”) with
triethylentetramine? I suggest you look at this paper (Accelerated co-precipitation of lead, zinc and
copper by carbon dioxide bubbling in alkaline municipal solid waste incinerator (MSWI) fly ash wash
water. By Wang et al.RSC Adv., 2016, 6, 20173) which explains mechanism of carbonation on page
- You will see that carbonation happens thanks to HCO3- which is produced when CO2 meets
OH- which in your case are supplied by the aqueous phase. Therefore, I would think that your sulfate
ions are not bonded to TETA but are free floating in the aqueous phase. I suggest you have a good
think about the mechanism of the carbonation and describe it in the manuscript. And include
refernce to carbonation process with CO2 into introduction.
ANSWER: Corrected and added.

Reviewer 2 Report
In this study, authors have applied pressure leaching technology for recovery of lead from solid residues of copper industry using triethylenetetramine as a leaching agent. Authors have applied multiple techniques for characterization solid residues ad well as precipitated products. This study may add knowledge on pyro- and hydrometallurgical-based methods for extraction and recovery lead from lead-containing waste materials. The reviewer has following comments that author should consider to improve the manuscript.
General comments:
Introduction: Author should clearly highlight what is the novelty of this study with respect to the existing knowledge on the research topic in literature.
Write a paragraph on implications of this study including scale-up of the process to use in industrial scale. Also, what future studies should be carried out to improve knowledge and further advancement of the technology.
Overall, the reported process looks complex which involves high pressure and temperature as well as numerous chemical reagents including inorganic acid and gases. The reviewer is wondering whether the developed method would be environmentally friendly and economically feasible. Could you list out the potential limitations of this technology. Also, there would be a lot of wastewater produced at various steps of the process. What is authors suggestion to manage this.
What are potential industrial applications of the precipitated lead given the purity level £76%.
Minor comments:
Line 89: “Ph was monitored…”Is it pH?
Line 117: “6 up to 20%.” Is it wt%?
Tables 1 - 4: In the footnote, expand all abbreviations to improve understanding.
Author Response
„In this study, authors have applied pressure leaching technology for recovery of lead from solid residues of copper industry using triethylenetetramine as a leaching agent. Authors have applied multiple techniques for characterization solid residues ad well as precipitated products. This study may add knowledge on pyro- and hydrometallurgical-based methods for extraction and recovery lead from lead-containing waste materials. The reviewer has following comments that author should consider to improve the manuscript.”
General comments:
Introduction: Author should clearly highlight what is the novelty of this study with respect to the existing knowledge on the research topic in literature.
ANSWER: The novelty is recycling of amine solution for lead extraction. A sentence concerning perspectives for obtained product was added in the abstract.
Write a paragraph on implications of this study including scale-up of the process to use in industrial scale. Also, what future studies should be carried out to improve knowledge and further advancement of the technology.
ANSWER: We were hasitating whether add this information or not. At this moment discussions and negtiations are made for perspective scale up with one of the international company. We would like to stay clear in this subject for some time.
Overall, the reported process looks complex which involves high pressure and temperature as well as numerous chemical reagents including inorganic acid and gases. The reviewer is wondering whether the developed method would be environmentally friendly and economically feasible. Could you list out the potential limitations of this technology. Also, there would be a lot of wastewater produced at various steps of the process. What is authors suggestion to manage this.
ANSWER: Methodology presented here deals only with lead sulfate extraction with amine. All initial operations like pressure leaching was used to make materials pretreatment and produce „real residue” (from real metal concentrates). Inorganic acid used in is 1mol/L sulfuric acid solution
What are potential industrial applications of the precipitated lead given the purity level £76%.
ANSWER: That’s not purity of lead. The product is lead carbonate, which is very attractive materials for smelting. Much better than lead sulfate as it does not increase smelting temperaturÄ™. Additionally, this material has high purity so there’s no risk to introduce other metals to the furnace.
Minor comments:
Line 89: “Ph was monitored…”Is it pH?
ANSWER: Corrected
Line 117: “6 up to 20%.” Is it wt%?
ANSWER: Yes. Corrected.
Tables 1 - 4: In the footnote, expand all abbreviations to improve understanding.
ANSWER: Abbreviations were given in „Materials and methods” sections to have better understanding.
Round 2
Reviewer 1 Report
attached

Reviewer 2 Report
No more comments.
Author Response
There was no comments from this Reviewer
This manuscript is a resubmission of an earlier submission. The following is a list of the peer review reports and author responses from that submission.
Round 1
Reviewer 1 Report
The article requires careful revision before it is ready for publication.
- The title of the article is too broad; it should be made more specific according to your work.
- The introduction of your work poorly reflects the relevance and novelty of the research, there is absolutely no information about the developments of other scientific groups in the field of lead extraction from solid waste. The purpose of the paper should be from a new paragraph and clearly described, now it gets lost in the text and poorly described. When writing the introduction, it is recommended to use references in highly rated journals of leading scientific publishers.
- The abstract should clearly reflect the result of your work in the article. Now it is similar to the text from the introduction.
- In the Materials and Methods section, all equipment should be described as follows: (model, company, city, state, country).
- Why was a time of 1 h and a temperature of 60 deg chosen to perform lead extraction using an amine?
- What is the basis of your proposed mechanism for lead extraction with triethyltetraamine?
- What formula was used to calculate the degree of lead extraction in the extract? Is your data reproducible?
- It is necessary to expand the experimental base in the field of leaching and extraction.
- All the results obtained are poorly described and require a great deal of attention to edit and write.
- You need to check the text for the English language: articles, grammar, punctuation, etc.
Reviewer 2 Report
In this manuscript, Ciszewski et al. reported findings on “Lead recovery from pressure leaching residues”. The proposed method may be useful for recovery of valuable metals from waste materials. The reviewer has raised a few questions on this study.
The “Results and Discussion” section looks like only results are reported. No in-depth discussion on the reported results with literature.
Is the proposed process economically feasible.
What are the potential limitations of the proposed method, and how to address them.
What are the specific reason for use of “triethylenetetramine”. Are there any alternative chemical reagents can be used.
In stead of H2SO4, can other inorganic acids be used.
Table 1: In the footnote, explain about the sample identity, i.e., OT8 – OT114. Also include the unit (%).
Figs. 4 and 5: Clearly mention the identity (compound name) of the XRD peaks, also give literature references for the peaks identified as PbSO4, Pb-jarosite.
In addition to XRD, it would be good if authors could provide SEM images (if available) of the raw minerals, acid treated residues and precipitated products to show the differences of microstructure among these materials.
Reviewer 3 Report
Introduction need to be improved. I think more detail info should be presented.
In reactions authors have to use lower and upper index for numbers in chemical compounds or ions.
I recommend to not use old units for concentration M (older unit mark) but mol dm-3 (mol pre decimeter cubid) according to IUAPC nomenclature.
Line 123 duration of decomposition experiment for 20 hours at 90°C. It is long time at high temperature. What about evaporation and acid concentration changes during this long time? Did you watch it and adjusted concentration of acid?
Particle size of solid material used for leaching have to be defined in materials and methods. Did you use only one granularity, or did you study also the effect of particles size on the efficiency? I recommend to add in article the effect of particles size on the process.
Extraction of lead sulfate with amine was carried out for 1h at 60°C with the solid:liquid ratio 0.2. Why did you use these conditions? Did you study effect of solid:liquid ratio on extraction process? Comparison of different ratios could improve you article.
There is not described in details the process of extraction with Triethylenetetramine as well as carbonation and gypsum production.
Reviewer 4 Report
Manuscript ID: minerals-1059228
Title: Lead recovery from pressure leaching residues
Authors: Mateusz Ciszewski et al.
Abstract. Authors must write full formulas of plumbojarosite, lead sulfate and triethylenetetramine.
Line 32-38. What’s about Chile, China, Peru, Russia experience at hydrometallurgical methods?
Line 46. Authors must write full formulas of hematite, iron sulfate, jarosites.
Line 49-52. Authors should use subscript symbols in reactions 1-4.
Line 56-60. Authors must write full formulas of acids and minerals.
Line 63. Institute of Non-Ferrous Metals. What’s the country (Poland?) and Authors names?
The introduction is so small. Authors must describe in more detail general sentences (Lines 54-55) and re-write all text in this section.
Line 67. What’s mining facilities? Add name and country.
Line 77. What’s sulfuric acid concentration? What’s the solid/liquid ratio? Whats the particle size distribution of samples OT8-OT14? What’s mean OT?
Line 101. Authors must write full formula of anglesite.
Figure 1. Not all peaks are signed, which of the samples is declared as the original one? In Table 1, OT8-OT12 samples differ significantly in chemical composition from OT13-OT14 in terms of the silicon, lead and sulfur content. It is necessary to give XRD of two different types of samples.
It is necessary to add figures with metals extraction degree or minerals decomposition depending on the concentration of sulfuric acid and the duration of leaching for each sample.
What is the rotation speed and duration of the process? What is the foaming agent, its concentration? What is flotation methods? What are the parameters of the product yield, the extraction of lead into concentrate, the efficiency of flotation?
Figure 4. Authors must write the full formula of Cerrusite.
Line 160-168. Authors must add chemical reactions and technological parameters of this process.
Table 3. The sum of the elements is not equal to 100 wt. %
The references do not match the style of the Minerals. No links from 2018-2020.
Main conclusion:
The article is very poorly written. There is no detailed description of the leaching and flotation methods. There is no detailed description of solid samples. There are no figures for the extraction of the metals into the solution. The mechanism and kinetics of the leaching process have not been studied. There are no SEM pictures of the samples before and after leaching.
In this form, this article does not correspond to the Minerals level and should be rejected.